# META-DATASET: A DATASET OF DATASETS FOR LEARNING TO LEARN FROM FEW EXAMPLES

**Eleni Triantafillou**[*][†]**, Tyler Zhu**[†]**, Vincent Dumoulin**[†]**, Pascal Lamblin**[†]**, Utku Evci**[†]**,**
**Kelvin Xu**[‡][†]**, Ross Goroshin**[†]**, Carles Gelada**[†]**, Kevin Swersky**[†]**,**
**Pierre-Antoine Manzagol**[†] **& Hugo Larochelle**[†]
[*]University of Toronto and Vector Institute, [†]Google AI, [‡]University of California, Berkeley
Correspondence to: `eleni@cs.toronto.edu`

## ABSTRACT

Few-shot classification refers to learning a classifier for new classes given only a few examples. While a plethora of models have emerged to tackle it, we find the procedure and datasets that are used to assess their progress lacking. To address this limitation, we propose META-DATASET: a new benchmark for training and evaluating models that is large-scale, consists of diverse datasets, and presents more realistic tasks. We experiment with popular baselines and meta-learners on META-DATASET, along with a competitive method that we propose. We analyze performance as a function of various characteristics of test tasks and examine the models' ability to leverage diverse training sources for improving their generalization. We also propose a new set of baselines for quantifying the benefit of meta-learning in META-DATASET. Our extensive experimentation has uncovered important research challenges and we hope to inspire work in these directions.

## 1 INTRODUCTION

Few-shot learning refers to learning new concepts from few examples, an ability that humans naturally possess, but machines still lack. Improving on this aspect would lead to more efficient algorithms that can flexibly expand their knowledge without requiring large labeled datasets. We focus on few-shot classification: classifying unseen examples into one of $N$ new 'test' classes, given only a few reference examples of each. Recent progress in this direction has been made by considering a meta-problem: though we are not interested in learning about any training class in particular, we can exploit the training classes for the purpose of *learning to learn new classes from few examples*, thus acquiring a learning procedure that can be directly applied to new few-shot learning problems too.

This intuition has inspired numerous models of increasing complexity (see Related Work for some examples). However, we believe that the commonly-used setup for measuring success in this direction is lacking. Specifically, two datasets have emerged as *de facto* benchmarks for few-shot learning: Omniglot (Lake et al., 2015), and *mini*-ImageNet (Vinyals et al., 2016), and we believe that both of them are approaching their limit in terms of allowing one to discriminate between the merits of different approaches. Omniglot is a dataset of 1623 handwritten characters from 50 different alphabets and contains 20 examples per class (character). Most recent methods obtain very high accuracy on Omniglot, rendering the comparisons between them mostly uninformative. *mini*-ImageNet is formed out of 100 ImageNet (Russakovsky et al., 2015) classes (64/16/20 for train/validation/test) and contains 600 examples per class. Albeit harder than Omniglot, it has the same property that most recent methods trained on it present similar accuracy when controlling for model capacity. We advocate that a more challenging and realistic benchmark is required for further progress in this area.

More specifically, current benchmarks: 1) Consider homogeneous learning tasks. In contrast, real-life learning experiences are heterogeneous: they vary in terms of the number of classes and examples per class, and are unbalanced. 2) Measure only within-dataset generalization. However, we are eventually after models that can generalize to entirely new distributions (e.g., datasets). 3) Ignore the relationships between classes when forming episodes. Specifically, the coarse-grained classification of dogs and chairs may present different difficulties than the fine-grained classification of dog breeds, and current benchmarks do not establish a distinction between the two.

META-DATASET aims to improve upon previous benchmarks in the above directions: it is significantly larger-scale and is comprised of multiple datasets of diverse data distributions; its task creation is informed by class structure for ImageNet and Omniglot; it introduces realistic class imbalance; and it varies the number of classes in each task and the size of the training set, thus testing the robustness of models across the spectrum from very-low-shot learning onwards.

The main contributions of this work are: 1) A more realistic, large-scale and diverse environment for training and testing few-shot learners. 2) Experimental evaluation of popular models, and a new set of baselines combining inference algorithms of meta-learners with non-episodic training. 3) Analyses of whether different models benefit from more data, heterogeneous training sources, pre-trained weights, and meta-training. 4) A novel meta-learner that performs strongly on META-DATASET.

## 2 FEW-SHOT CLASSIFICATION: TASK FORMULATION AND APPROACHES

**Task Formulation**   The end-goal of few-shot classification is to produce a model which, given a new learning *episode* with $N$ classes and a few labeled examples ($k_c$ per class, $c \in 1, \ldots, N$), is able to generalize to unseen examples for that episode. In other words, the model learns from a training *(support)* set $\mathcal{S} = \{(\mathbf{x}_1, y_1), (\mathbf{x}_2, y_2), \ldots, (\mathbf{x}_K, y_K)\}$ (with $K = \sum_c k_c$) and is evaluated on a held-out test *(query)* set $\mathcal{Q} = \{(\mathbf{x}_1^*, y_1^*), (\mathbf{x}_2^*, y_2^*), \ldots, (\mathbf{x}_T^*, y_T^*)\}$. Each example $(\mathbf{x}, y)$ is formed of an input vector $\mathbf{x} \in \mathbb{R}^D$ and a class label $y \in \{1, \ldots, N\}$. Episodes with balanced training sets (i.e., $k_c = k, \ \forall c$) are usually described as '$N$-way, $k$-shot' episodes. Evaluation episodes are constructed by sampling their $N$ classes from a larger set $\mathcal{C}_{test}$ of classes and sampling the desired number of examples per class.

A disjoint set $\mathcal{C}_{train}$ of classes is available to train the model; note that this notion of training is distinct from the training that occurs within a few-shot learning episode. Few-shot learning does not prescribe a specific procedure for exploiting $\mathcal{C}_{train}$, but a common approach matches the conditions in which the model is trained and evaluated (Vinyals et al., 2016). In other words, training often (but not always) proceeds in an episodic fashion. Some authors use *training* and *testing* to refer to what happens *within* any given episode, and *meta-training* and *meta-testing* to refer to using $\mathcal{C}_{train}$ to turn the model into a learner capable of fast adaptation and $\mathcal{C}_{test}$ for evaluating its success to learn using few shots, respectively. This nomenclature highlights the *meta-learning* perspective alluded to earlier, but to avoid confusion we will adopt another common nomenclature and refer to the training and test sets of an episode as the *support* and *query* sets and to the process of learning from $\mathcal{C}_{train}$ simply as training. We use the term 'meta-learner' to describe a model that is trained episodically, i.e., learns to learn across multiple tasks that are sampled from the training set $\mathcal{C}_{train}$.

**Non-episodic Approaches to Few-shot Classification**   A natural non-episodic approach simply trains a classifier over all of the training classes $\mathcal{C}_{train}$ at once, which can be parameterized by a neural network with a linear layer on top with one output unit per class. After training, this neural network is used as an embedding function $g$ that maps images into a meaningful representation space. The hope of using this model for few-shot learning is that this representation space is useful even for examples of classes that were not included in training. It would then remain to define an algorithm for performing few-shot classification on top of these representations of the images of a task. We consider two choices for this algorithm, yielding the '$k$-NN' and 'Finetune' variants of this baseline.

Given a test episode, the '$k$-NN' baseline classifies each query example as the class that its 'closest' support example belongs to. Closeness is measured by either Euclidean or cosine distance in the learned embedding space; a choice that we treat as a hyperparameter. On the other hand, the 'Finetune' baseline uses the support set of the given test episode to train a new 'output layer' on top of the embeddings $g$, and optionally finetune those embedding too (another hyperparameter), for the purpose of classifying between the $N$ new classes of the associated task.

A variant of the 'Finetune' baseline has recently become popular: Baseline++ (Chen et al., 2019), originally inspired by Gidaris & Komodakis (2018); Qi et al. (2018). It uses a 'cosine classifier' as the final layer ($\ell^2$-normalizing embeddings and weights before taking the dot product), both during the non-episodic training phase, and for evaluation on test episodes. We incorporate this idea in our codebase by adding a hyperparameter that optionally enables using a cosine classifier for the '$k$-NN' (training only) and 'Finetune' (both phases) baselines.

**Meta-Learners for Few-shot Classification**    In the episodic setting, models are trained end-to-end for the purpose of learning to build classifiers from a few examples. We choose to experiment with Matching Networks (Vinyals et al., 2016), Relation Networks (Sung et al., 2018), Prototypical Networks (Snell et al., 2017) and Model Agnostic Meta-Learning (MAML, Finn et al., 2017) since they cover a diverse set of approaches to few-shot learning. We also introduce a novel meta-learner which is inspired by the last two models.

In each training episode, episodic models compute for each query example $\mathbf{x}^* \in \mathcal{Q}$, the distribution for its label $p(y^*|\mathbf{x}^*, \mathcal{S})$ conditioned on the support set $\mathcal{S}$ and allow to train this differentiably-parameterized conditional distribution end-to-end via gradient descent. The different models are distinguished by the manner in which this conditioning on the support set is realized. In all cases, the performance on the query set drives the update of the meta-learner's weights, which include (and sometimes consist only of) the embedding weights. We briefly describe each method below.

**Prototypical Networks**    Prototypical Networks construct a prototype for each class and then classify each query example as the class whose prototype is 'nearest' to it under Euclidean distance. More concretely, the probability that a query example $\mathbf{x}^*$ belongs to class $k$ is defined as:

$$p(y^* = k|\mathbf{x}^*, \mathcal{S}) = \frac{\exp(-||g(\mathbf{x}^*) - \mathbf{c}_k||_2^2)}{\sum_{k' \in \{1,...,N\}} \exp(-||g(\mathbf{x}^*) - \mathbf{c}_{k'}||_2^2)}$$

where $\mathbf{c}_k$ is the 'prototype' for class $k$: the average of the embeddings of class $k$'s support examples.

**Matching Networks**    Matching Networks (in their simplest form) label each query example as a (cosine) distance-weighted linear combination of the support labels:

$$p(y^* = k|\mathbf{x}^*, \mathcal{S}) = \sum_{i=1}^{|\mathcal{S}|} \alpha(\mathbf{x}^*, \mathbf{x}_i) \mathbf{1}_{y_i = k},$$

where $\mathbf{1}_A$ is the indicator function and $\alpha(\mathbf{x}^*, \mathbf{x}_i)$ is the cosine similarity between $g(\mathbf{x}^*)$ and $g(\mathbf{x}_i)$, softmax-normalized over all support examples $\mathbf{x}_i$, where $1 \leq i \leq |\mathcal{S}|$.

**Relation Networks**    Relation Networks are comprised of an embedding function $g$ as usual, and a 'relation module' parameterized by some additional neural network layers. They first embed each support and query using $g$ and create a prototype $p_c$ for each class $c$ by averaging its support embeddings. Each prototype $p_c$ is concatenated with each embedded query and fed through the relation module which outputs a number in $[0, 1]$ representing the predicted probability that that query belongs to class $c$. The query loss is then defined as the mean square error of that prediction compared to the (binary) ground truth. Both $g$ and the relation module are trained to minimize this loss.

**MAML**    MAML uses a linear layer parametrized by $\mathbf{W}$ and $\mathbf{b}$ on top of the embedding function $g(\cdot; \theta)$ and classifies a query example as

$$p(y^*|\mathbf{x}^*, \mathcal{S}) = \text{softmax}(\mathbf{b}' + \mathbf{W}' g(\mathbf{x}^*; \theta')),$$

where the output layer parameters $\mathbf{W}'$ and $\mathbf{b}'$ and the embedding function parameters $\theta'$ are obtained by performing a small number of within-episode training steps on the support set $S$, starting from initial parameter values $(\mathbf{b}, \mathbf{W}, \theta)$. The model is trained by backpropagating the query set loss through the within-episode gradient descent procedure and into $(\mathbf{b}, \mathbf{W}, \theta)$. This normally requires computing second-order gradients, which can be expensive to obtain (both in terms of time and memory). For this reason, an approximation is often used whereby gradients of the within-episode descent steps are ignored. This variant is referred to as first-order MAML (fo-MAML) and was used in our experiments. We did attempt to use the full-order version, but found it to be impractically expensive (e.g., it caused frequent out-of-memory problems).

Moreover, since in our setting the number of ways varies between episodes, $\mathbf{b}, \mathbf{W}$ are set to zero and are not trained (i.e., $\mathbf{b}', \mathbf{W}'$ are the result of within-episode gradient descent initialized at 0), leaving only $\theta$ to be trained. In other words, MAML focuses on learning the within-episode initialization $\theta$ of the embedding network so that it can be rapidly adapted for a new task.

**Introducing Proto-MAML** We introduce a novel meta-learner that combines the complementary strengths of Prototypical Networks and MAML: the former's simple inductive bias that is evidently effective for very-few-shot learning, and the latter's flexible adaptation mechanism.

As explained by Snell et al. (2017), Prototypical Networks can be re-interpreted as a linear classifier applied to a learned representation $g(\mathbf{x})$. The use of a squared Euclidean distance means that output logits are expressed as

$$-||g(\mathbf{x}^*) - \mathbf{c}_k||^2 = -g(\mathbf{x}^*)^T g(\mathbf{x}^*) + 2\mathbf{c}_k^T g(\mathbf{x}^*) - \mathbf{c}_k^T \mathbf{c}_k = 2\mathbf{c}_k^T g(\mathbf{x}^*) - ||\mathbf{c}_k||^2 + constant$$

where $constant$ is a class-independent scalar which can be ignored, as it leaves output probabilities unchanged. The $k$-th unit of the equivalent linear layer therefore has weights $\mathbf{W}_{k,.} = 2\mathbf{c}_k$ and biases $b_k = -||\mathbf{c}_k||^2$, which are both differentiable with respect to $\theta$ as they are a function of $g(\cdot; \theta)$.

We refer to (fo-)Proto-MAML as the (fo-)MAML model where the task-specific linear layer of each episode is initialized from the Prototypical Network-equivalent weights and bias defined above and subsequently optimized as usual on the given support set. When computing the update for $\theta$, we allow gradients to flow through the Prototypical Network-equivalent linear layer initialization. We show that this simple modification significantly helps the optimization of this model and outperforms vanilla fo-MAML by a large margin on META-DATASET.

## 3 META-DATASET: A NEW FEW-SHOT CLASSIFICATION BENCHMARK

META-DATASET aims to offer an environment for measuring progress in realistic few-shot classification tasks. Our approach is twofold: 1) changing the data and 2) changing the formulation of the task (i.e., how episodes are generated). The following sections describe these modifications in detail. The code is open source and publicly available[1].

### 3.1 META-DATASET'S DATA

META-DATASET's data is much larger in size than any previous benchmark, and is comprised of *multiple existing datasets*. This invites research into how diverse sources of data can be exploited by a meta-learner, and allows us to evaluate a more challenging generalization problem, to new datasets altogether. Specifically, META-DATASET leverages data from the following 10 datasets: ILSVRC-2012 (ImageNet, Russakovsky et al., 2015), Omniglot (Lake et al., 2015), Aircraft (Maji et al., 2013), CUB-200-2011 (Birds, Wah et al., 2011), Describable Textures (Cimpoi et al., 2014), Quick Draw (Jongejan et al., 2016), Fungi (Schroeder & Cui, 2018), VGG Flower (Nilsback & Zisserman, 2008), Traffic Signs (Houben et al., 2013) and MSCOCO (Lin et al., 2014). These datasets were chosen because they are free and easy to obtain, span a variety of visual concepts (natural and human-made) and vary in how fine-grained the class definition is. More information about each of these datasets is provided in the Appendix.

To ensure that episodes correspond to realistic classification problems, each episode generated in META-DATASET uses classes from a single dataset. Moreover, two of these datasets, Traffic Signs and MSCOCO, are fully reserved for evaluation, meaning that no classes from them participate in the training set. The remaining ones contribute some classes to each of the training, validation and test splits of classes, roughly with 70% / 15% / 15% proportions. Two of these datasets, ImageNet and Omniglot, possess a class hierarchy that we exploit in META-DATASET. For each dataset, the composition of splits is available online[2].

**ImageNet** ImageNet is comprised of 82,115 'synsets', i.e., concepts of the WordNet ontology, and it provides 'is-a' relationships for its synsets, thus defining a DAG over them. META-DATASET uses the 1K synsets that were chosen for the ILSVRC 2012 classification challenge and defines a new class split for it and a novel procedure for sampling classes from it for episode creation, both informed by its class hierarchy.

Specifically, we construct a sub-graph of the overall DAG whose leaves are the 1K classes of ILSVRC-2012. We then 'cut' this sub-graph into three pieces, for the training, validation, and test splits,

---

[1]github.com/google-research/meta-dataset
[2]github.com/google-research/meta-dataset/tree/master/meta_dataset/dataset_conversion/splits

such that there is no overlap between the leaves of any of these pieces. For this, we selected the synsets 'carnivore' and 'device' as the roots of the validation and test sub-graphs, respectively. The leaves that are reachable from 'carnivore' and 'device' form the sets of the validation and test classes, respectively. All of the remaining leaves constitute the training classes. This method of splitting ensures that the training classes are semantically different from the test classes. We end up with 712 training, 158 validation and 130 test classes, roughly adhering to the standard 70 / 15 / 15 (%) splits.

**Omniglot**   This dataset is one of the established benchmarks for few-shot classification as mentioned earlier. However, contrary to the common setup that flattens and ignores its two-level hierarchy of alphabets and characters, we allow it to influence the episode class selection in META-DATASET, yielding finer-grained tasks. We also use the original splits proposed in Lake et al. (2015): (all characters of) the 'background' and 'evaluation' alphabets are used for training and testing, respectively. However, we reserve the 5 smallest alphabets from the 'background' set for validation.

## 3.2   EPISODE SAMPLING

In this section we outline META-DATASET's algorithm for sampling episodes, featuring hierarchically-aware procedures for sampling classes of ImageNet and Omniglot, and an algorithm that yields realistically imbalanced episodes of variable shots and ways. The steps for sampling an episode for a given split are: Step 0) uniformly sample a dataset $\mathcal{D}$, Step 1) sample a set of classes $\mathcal{C}$ from the classes of $\mathcal{D}$ assigned to the requested split, and Step 2) sample support and query examples from $\mathcal{C}$.

**Step 1: Sampling the episode's class set**   This procedure differs depending on which dataset is chosen. For datasets without a known class organization, we sample the 'way' uniformly from the range $[5, \text{MAX-CLASSES}]$, where MAX-CLASSES is either $50$ or as many as there are available. Then we sample 'way' many classes uniformly at random from the requested class split of the given dataset. ImageNet and Omniglot use class-structure-aware procedures outlined below.

**ImageNet class sampling**   We adopt a hierarchy-aware sampling procedure: First, we sample an internal (non-leaf) node uniformly from the DAG of the given split. The chosen set of classes is then the set of leaves spanned by that node (or a random subset of it, if more than 50). We prevent nodes that are too close to the root to be selected as the internal node, as explained in more detail in the Appendix. This procedure enables the creation of tasks of varying degrees of fine-grainedness: the larger the height of the internal node, the more coarse-grained the resulting episode.

**Omniglot class sampling**   We sample classes from Omniglot by first sampling an alphabet uniformly at random from the chosen split of alphabets (train, validation or test). Then, the 'way' of the episode is sampled uniformly at random using the same restrictions as for the rest of the datasets, but taking care not to sample a larger number than the number of characters that belong to the chosen alphabet. Finally, the prescribed number of characters of that alphabet are randomly sampled. This ensures that each episode presents a within-alphabet fine-grained classification.

**Step 2: Sampling the episode's examples**   Having already selected a set of classes, the choice of the examples from them that will populate an episode can be broken down into three steps. We provide a high-level description here and elaborate in the Appendix with the accompanying formulas.

**Step 2a: Compute the query set size**   The query set is class-balanced, reflecting the fact that we care equally to perform well on all classes of an episode. The number of query images *per class* is set to a number such that all chosen classes have enough images to contribute that number and still remain with roughly half on their images to possibly add to the support set (in a later step). This number is capped to 10 images per class.

**Step 2b: Compute the support set size**   We allow each chosen class to contribute to the support set at most 100 of its remaining examples (i.e., excluding the ones added to the query set). We multiply this remaining number by a scalar sampled uniformly from the interval $(0, 1]$ to enable the potential generation of 'few-shot' episodes even when multiple images are available, as we are also interested in studying that end of the spectrum. We do enforce, however, that each chosen class has a budget for at least one image in the support set, and we cap the total support set size to 500 examples.

**Step 2c: Compute the shot of each class** We now discuss how to distribute the total support set size chosen above across the participating classes. The un-normalized proportion of the support set that will be occupied by a given chosen class is a noisy version of the total number of images of that class in the dataset. This design choice is made in the hopes of obtaining realistic class ratios, under the hypothesis that the dataset class statistics are a reasonable approximation of the real-world statistics of appearances of the corresponding classes. We ensure that each class has at least one image in the support set and distribute the rest according to the above rule.

After these steps, we complete the episode creation process by choosing the prescribed number of examples of each chosen class uniformly at random to populate the support and query sets.

## 4 RELATED WORK

In this work we evaluate four meta-learners on META-DATASET that we believe capture a good diversity of well-established models. Evaluating other few-shot classifiers on META-DATASET is beyond the scope of this paper, but we discuss some additional related models below.

Similarly to MAML, some train a meta-learner for quick adaptation to new tasks (Ravi & Larochelle, 2017; Munkhdalai & Yu, 2017; Rusu et al., 2019; Yoon et al., 2018). Others relate to Prototypical Networks by learning a representation on which differentiable training can be performed on some form of classifier (Bertinetto et al., 2019; Gidaris & Komodakis, 2018; Oreshkin et al., 2018). Others relate to Matching Networks in that they perform comparisons between pairs of support and query examples, using either a graph neural network (Satorras & Estrach, 2018) or an attention mechanism (Mishra et al., 2018). Finally, some make use of memory-augmented recurrent networks (Santoro et al., 2016), some learn to perform data augmentation (Hariharan & Girshick, 2017; Wang et al., 2018) in a low-shot learning setting, and some learn to predict the parameters of a large-shot classifier from the parameters learned in a few-shot setting (Wang & Hebert, 2016; Wang et al., 2017). Of relevance to Proto-MAML is MAML++ (Antoniou et al., 2019), which consists of a collection of adjustments to MAML, such as multiple meta-trained inner loop learning rates and derivative-order annealing. Proto-MAML instead modifies the output weight initialization scheme and could be combined with those adjustments.

Finally, META-DATASET relates to other recent image classification benchmarks. The CVPR 2017 Visual Domain Decathlon Challenge trains a model on 10 different datasets, many of which are included in our benchmark, and measures its ability to generalize to held-out examples for those same datasets but does not measure generalization to new classes (or datasets). Hariharan & Girshick (2017) propose a benchmark where a model is given abundant data from certain *base* ImageNet classes and is tested on few-shot learning *novel* ImageNet classes in a way that doesn't compromise its knowledge of the base classes. Wang et al. (2018) build upon that benchmark and propose a new evaluation protocol for it. Chen et al. (2019) investigate fine-grained few-shot classification using the CUB dataset (Wah et al., 2011, also featured in our benchmark) and cross-domain transfer between *mini*-ImageNet and CUB. Larger-scale few-shot classification benchmarks were also proposed using CIFAR-100 (Krizhevsky et al., 2009; Bertinetto et al., 2019; Oreshkin et al., 2018), *tiered*-ImageNet (Ren et al., 2018), and ImageNet-21k (Dhillon et al., 2019). Compared to these, META-DATASET contains the largest set of diverse datasets in the context of few-shot learning and is additionally accompanied by an algorithm for creating learning scenarios from that data that we advocate are more realistic than the previous ones.

## 5 EXPERIMENTS

**Training procedure** META-DATASET does not prescribe a procedure for learning from the training data. In these experiments, keeping with the spirit of matching training and testing conditions, we trained the meta-learners via training episodes sampled using the same algorithm as we used for META-DATASET's evaluation episodes, described above. The choice of the dataset from which to sample the next episode was random uniform. The non-episodic baselines are trained to solve the large classification problem that results from 'concatenating' the training classes of all datasets.

**Validation** Another design choice was to perform validation on (the validation split of) ImageNet only, ignoring the validation sets of the other datasets. The rationale behind this choice is that the

Table 1: Few-shot classification results on META-DATASET using models **trained on ILSVRC-2012 only (top)** and **trained on all datasets (bottom)**.

| Test Source | $k$-NN | Finetune | MatchingNet | ProtoNet | fo-MAML | RelationNet | fo-Proto-MAML |
|---|---|---|---|---|---|---|---|
| ILSVRC | 41.03 | 45.78 | 45.00 | **50.50** | 45.51 | 34.69 | **49.53** |
| Omniglot | 37.07 | 60.85 | 52.27 | 59.98 | 55.55 | 45.35 | **63.37** |
| Aircraft | 46.81 | **68.69** | 48.97 | 53.10 | 56.24 | 40.73 | 55.95 |
| Birds | 50.13 | 57.31 | 62.21 | **68.79** | 63.61 | 49.51 | **68.66** |
| Textures | 66.36 | **69.05** | 64.15 | 66.56 | **68.04** | 52.97 | 66.49 |
| Quick Draw | 32.06 | 42.60 | 42.87 | 48.96 | 43.96 | 43.30 | **51.52** |
| Fungi | 36.16 | 38.20 | 33.97 | **39.71** | 32.10 | 30.55 | **39.96** |
| VGG Flower | 83.10 | 85.51 | 80.13 | 85.27 | 81.74 | 68.76 | **87.15** |
| Traffic Signs | 44.59 | **66.79** | 47.80 | 47.12 | 50.93 | 33.67 | 48.83 |
| MSCOCO | 30.38 | 34.86 | 34.99 | 41.00 | 35.30 | 29.15 | **43.74** |
| **Avg. rank** | 5.7 | 2.9 | 4.65 | 2.65 | 3.7 | 6.55 | **1.85** |

| Test Source | $k$-NN | Finetune | MatchingNet | ProtoNet | fo-MAML | RelationNet | fo-Proto-MAML |
|---|---|---|---|---|---|---|---|
| ILSVRC | 38.55 | 43.08 | 36.08 | 44.50 | 37.83 | 30.89 | **46.52** |
| Omniglot | 74.60 | 71.11 | 78.25 | 79.56 | 83.92 | **86.57** | 82.69 |
| Aircraft | 64.98 | 72.03 | 69.17 | 71.14 | **76.41** | 69.71 | 75.23 |
| Birds | 66.35 | 59.82 | 56.40 | 67.01 | 62.43 | 54.14 | **69.88** |
| Textures | 63.58 | **69.14** | 61.80 | 65.18 | 64.16 | 56.56 | 68.25 |
| Quick Draw | 44.88 | 47.05 | 60.81 | 64.88 | 59.73 | 61.75 | **66.84** |
| Fungi | 37.12 | 38.16 | 33.70 | 40.26 | 33.54 | 32.56 | **41.99** |
| VGG Flower | 83.47 | 85.28 | 81.90 | 86.85 | 79.94 | 76.08 | **88.72** |
| Traffic Signs | 40.11 | **66.74** | 55.57 | 46.48 | 42.91 | 37.48 | 52.42 |
| MSCOCO | 29.55 | 35.17 | 28.79 | 39.87 | 29.37 | 27.41 | **41.74** |
| **Avg. rank** | 5.05 | 3.6 | 4.95 | 2.85 | 4.25 | 5.8 | **1.5** |

performance on ImageNet has been known to be a good proxy for the performance on different datasets. We used this validation performance to select our hyperparameters, including backbone architectures, image resolutions and model-specific ones. We describe these further in the Appendix.

**Pre-training** We gave each meta-learner the opportunity to initialize its embedding function from the embedding weights to which the $k$-NN Baseline model trained on ImageNet converged to. We treated the choice of starting from scratch or starting from this initialization as a hyperparameter. For a fair comparison with the baselines, we allowed the non-episodic models to start from this initialization too. This is especially important for the baselines in the case of training on all datasets since it offers the opportunity to start from ImageNet-pretrained weights.

**Main results** Table 1 displays the accuracy of each model on the test set of each dataset, after they were trained on ImageNet-only or all datasets. Traffic Signs and MSCOCO are not used for training in either case, as they are reserved for evaluation. We propose to use the average (over the datasets) rank of each method as our metric for comparison, where smaller is better. A method receives rank 1 if it has the highest accuracy, rank 2 if it has the second highest, and so on. If two models share the best accuracy, they both get rank 1.5, and so on. We find that fo-Proto-MAML is the top-performer according to this metric, Prototypical Networks also perform strongly, and the Finetune Baseline notably presents a worthy opponent[3]. We include more detailed versions of these tables displaying confidence intervals and per-dataset ranks in the Appendix.

**Effect of training on all datasets instead of ImageNet only** It's interesting to examine whether training on (the training splits of) all datasets leads to improved generalizaton compared to training on (the training split of) ImageNet only. Specifically, while we might expect that training on more data helps improve generalization, it is an empirical question whether that still holds for heterogeneous data. We can examine this by comparing the performance of each model between the top and bottom sets of results of Table 1, corresponding to the two training sources (ImageNet only and all datasets, respectively). For convenience, Figure 1 visualizes this difference in a barplot. Notably, for Omniglot, Quick Draw and Aircraft we observe a substantial increase across the board from training on all

---

[3]We improved MAML's performance significantly in the time between the acceptance decision and publication, thanks to a suggestion to use a more aggressive inner-loop learning rate. In the Appendix, we present the older MAML results side by side with the new ones, and discuss the importance of this hyperparameter for MAML.

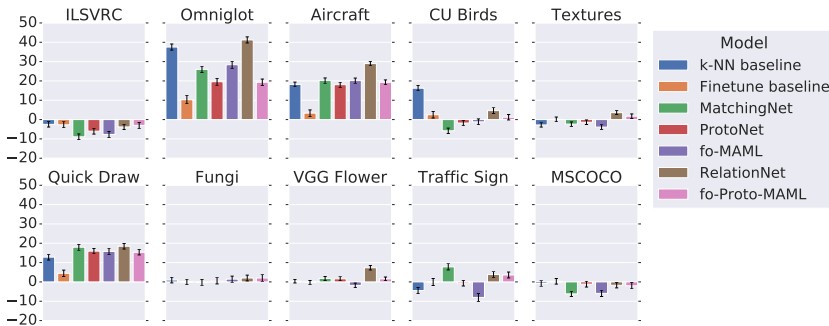

Figure 1: The performance difference on test datasets, when training on all datasets instead of ILSVRC only. A positive value indicates an improvement from all-dataset training.

sources. This is reasonable for datasets whose images are significantly different from ImageNet's: we indeed expect to gain a large benefit from training on some images from (the training classes of) these datasets. Interestingly though, on the remainder of the test sources, we don't observe a gain from all-dataset training. This result invites research into methods for exploiting heterogeneous data for generalization to unseen classes of diverse sources. Our experiments show that learning 'naively' across the training datasets (e.g., by picking the next dataset to use uniformly at random) does not automatically lead to that desired benefit in most cases.

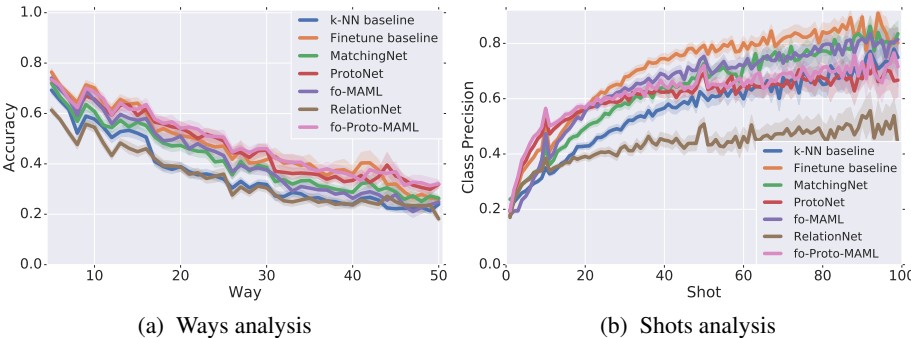

(a) Ways analysis

(b) Shots analysis

Figure 2: The effect of different ways and shots on test performance (w/ 95% confidence intervals) when training on ImageNet.

**Ways and shots analysis**   We further study the accuracy as a function of 'ways' (Figure 2a) and the class precision as a function of 'shots' (Figure 2b). As expected, we found that the difficulty increases as the way increases, and performance degrades. More examples per class, on the other hand, indeed make it easier to correctly classify that class. Interestingly, though, not all models benefit at the same rate from more data: Prototypical Networks and fo-Proto-MAML outshine other models in very-low-shot settings but saturate faster, whereas the Finetune baseline, Matching Networks, and fo-MAML improve at a higher rate when the shot increases. We draw the same conclusions when performing this analysis on all datasets, and include those plots in the Appendix. As discussed in the Appendix, we recommend including this analysis when reporting results on Meta-Dataset, aside from the main table. The rationale is that we're not only interested in performing well on average, but also in performing well under different specifications of test tasks.

**Effect of pre-training**   In Figures 3a and 3b, we quantify how beneficial it is to initialize the embedding network of meta-learners using the weights of the $k$-NN baseline pre-trained on ImageNet, as opposed to starting their episodic training from scratch. We find this procedure to often be beneficial, both for ImageNet-only training and for training on all datasets. It seems that this ImageNet-influenced initialization drives the meta-learner towards a solution which yields increased performance on natural image test datasets, especially ILSVRC, Birds, Fungi, Flowers and MSCOCO.

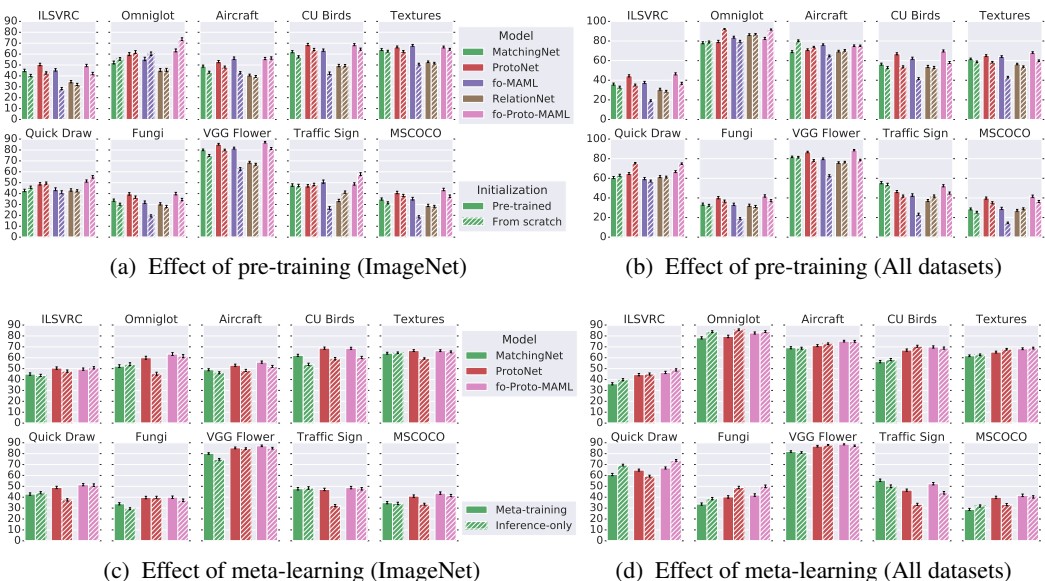

Figure 3: The effects of pre-training and meta-training (w/ 95% confidence intervals). (ImageNet) or (All datasets) is the training source.

Perhaps unsuspisingly, though, it underperforms on significantly different datasets such as Omniglot and Quick Draw. These findings show that, aside from the choice of the training data source(s) (e.g., ImageNet only or all datasets, as discussed above), the choice of the initialization scheme can also influence to an important degree the final solution and consequently the aptness of applying the resulting meta-learner to different data sources at test time. Finally, an interesting observation is that MAML seems to benefit the most from the pre-trained initialization, which may speak to the difficulty of optimization associated with that model.

**Effect of meta-training**   We propose to disentangle the inference algorithm of each meta-learner from the fact that it is meta-learned, to assess the benefit of meta-learning on META-DATASET. To this end, we propose a new set of baselines: 'Prototypical Networks Inference', 'Matching Networks Inference', and 'fo-Proto-MAML Inference', that are trained non-episodically but evaluated episodically (for validation and testing) using the inference algorithm of the respective meta-learner. This is possible for these meta-learners as they don't have any additional parameters aside from the embedding function that explicitly need to be learned episodically (as opposed to the relation module of Relation Networks, for example). We compare each Inference-only method to its corresponding meta-learner in Figures 3c and 3d. We find that these baselines are strong: when training on ImageNet only, we can usually observe a small benefit from meta-learning the embedding weights but this benefit often disappears when training on all datasets, in which case meta-learning sometimes actually hurts. We find this result very interesting and we believe it emphasizes the need for research on how to meta-learn across multiple diverse sources, an important challenge that META-DATASET puts forth.

**Fine-grainedness analysis**   We use ILVRC-2012 to investigate the hypothesis that finer-grained tasks are harder than coarse-grained ones. Our findings suggest that while the test sub-graph is not rich enough to exhibit any trend, the performance on the train sub-graph does seem to agree with this hypothesis. We include the experimental setup and results for this analysis in the Appendix.

## 6   CONCLUSION

We have introduced a new large-scale, diverse, and realistic environment for few-shot classification. We believe that our exploration of various models on META-DATASET has uncovered interesting

directions for future work pertaining to meta-learning across heterogeneous data: it remains unclear what is the best strategy for creating training episodes, the most appropriate validation creation and the most appropriate initialization. Current models don't always improve when trained on multiple sources and meta-learning is not always beneficial across datasets. Current models are also not robust to the amount of data in test episodes, each excelling in a different part of the spectrum. We believe that addressing these shortcomings consitutes an important research goal moving forward.

### AUTHOR CONTRIBUTIONS

Eleni, Hugo, and Kevin came up with the benchmark idea and requirements. Eleni developed the core of the project, and worked on the experiment design and management with Tyler and Kevin, as well as experiment analysis. Carles, Ross, Kelvin, Pascal, Vincent, and Tyler helped extend the benchmark by adding datasets. Eleni, Vincent, and Utku contributed the Prototypical Networks, Matching Networks, and Relation Networks implementations, respectively. Tyler implemented baselines, MAML (with Kevin) and Proto-MAML models, and updated the backbones to support them. Writing was mostly led by Eleni, with contributions by Hugo, Vincent, and Kevin and help from Tyler and Pascal for visualizations. Pascal and Pierre-Antoine worked on code organization, efficiency, and open-sourcing, Pascal and Vincent optimized the efficiency of the data input pipeline. Pierre-Antoine supervised the code development process and reviewed most of the changes, Hugo and Kevin supervised the overall direction of the research.

### ACKNOWLEDGMENTS

We would like to thank Chelsea Finn for fruitful discussions and advice on tuning fo-MAML and ensuring the correctness of implementation, as well as Zack Nado and Dan Moldovan for the initial dataset code that was adapted, and Cristina Vasconcelos for spotting an issue in the ranking of models. Finally, we'd like to thank John Bronskill for suggesting that we experiment with a larger inner-loop learning rate for MAML which indeed significantly improved our fo-MAML results on META-DATASET.

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

APPENDIX

### .1 RECOMMENDATION FOR REPORTING RESULTS ON META-DATASET

We recommend that future work on META-DATASET reports two sets of results:

1. The main tables storing the average (over 600 test episodes) accuracy of each method on each dataset, after it has been trained on ImageNet only and on All datasets, where the evaluation metric is the average rank. This corresponds to Table 1 in our case (or the more complete version in Table 2 in the Appendix).
2. The plots that measure robustness in variations of shots and ways. In our case these are Figures 2b and 2a in the main text for ImageNet-only training, and Figures 5b and 5a in the Appendix for the case of training on all datasets.

We propose to use both of these aspects to evaluate performance on META-DATASET: it is not only desirable to perform well on average, but also to perform well under different specifications of test tasks, as it is not realistic in general to assume that we will know in advance what setup (number of ways and shots) will be encountered at test time. Our final source code will include scripts for generating these plots and for automatically computing ranks given a table to help standardize the procedure for reporting results.

### .2 DETAILS OF META-DATASET'S SAMPLING ALGORITHM

We now provide a complete description of certain steps that were explained on a higher level in the main paper.

#### STEP 1: SAMPLING THE EPISODE'S CLASS SET

**ImageNet class sampling** The procedure we use for sampling classes for an ImageNet episode is the following. First, we sample a node uniformly at random from the set of 'eligible' nodes of the DAG structure corresponding to the specified split (train, validation or test). An internal node is 'eligible' for this selection if it spans at least 5 leaves, but no more than 392 leaves. The number 392 was chosen because it is the smallest number so that, collectively, all eligible internal nodes span all leaves in the DAG. Once an eligible node is selected, some of the leaves that it spans will constitute the classes of the episode. Specifically, if the number of those leaves is no greater than 50, we use all of them. Otherwise, we randomly choose 50 of them.

This procedure enables the creation of tasks of varying degrees of fine-grainedness. For instance, if the sampled internal node has a small height, the leaf classes that it spans will represent semantically-related concepts, thus posing a fine-grained classification task. As the height of the sampled node increases, we 'zoom out' to consider a broader scope from which we sample classes and the resulting episodes are more coarse-grained.

#### STEP 2: SAMPLING THE EPISODE'S EXAMPLES

**a) Computing the query set size** The query set is class-balanced, reflecting the fact that we care equally to perform well on all classes of an episode. The number of query images *per class* is computed as:

$$q = \min \left\{ 10, \left( \min_{c \in \mathcal{C}} \lfloor 0.5 * |Im(c)| \rfloor \right) \right\}$$

where $\mathcal{C}$ is the set of selected classes and $Im(c)$ denotes the set of images belonging to class $c$. The min over classes ensures that each class has at least $q$ images to add to the query set, thus allowing it to be class-balanced. The 0.5 multiplier ensures that enough images of each class will be available to add to the support set, and the minimum with 10 prevents the query set from being too large.

**b) Computing the support set size** We compute the total support set size as:

$$|\mathcal{S}| = \min \left\{ 500, \sum_{c \in \mathcal{C}} \lceil \beta \min\{100, |Im(c)| - q\} \rceil \right\}$$

where $\beta$ is a scalar sampled uniformly from interval $(0, 1]$. Intuitively, each class on average contributes either all its remaining examples (after placing $q$ of them in the query set) if there are less than 100 or 100 otherwise, to avoid having too large support sets. The multiplication with $\beta$ enables the potential generation of smaller support sets even when multiple images are available, since we are also interested in examining the very-low-shot end of the spectrum. The 'ceiling' operation ensures that each selected class will have at least one image in the support set. Finally, we cap the total support set size to 500.

**c) Computing the shot of each class**   We are now ready to compute the 'shot' of each class. Specifically, the proportion of the support set that will be devoted to class $c$ is computed as:

$$R_c = \frac{\exp(\alpha_c)|Im(c)|}{\sum\limits_{c' \in \mathcal{C}} \exp(\alpha'_c)|Im(c')|}$$

where $\alpha_c$ is sampled uniformly from the interval $[\log(0.5), \log(2))$. Intuitively, the un-normalized proportion of the support set that will be occupied by class $c$ is a noisy version of the total number of images of that class in the dataset $Im(c)$. This design choice is made in the hopes of obtaining realistic class ratios, under the hypothesis that the dataset class statistics are a reasonable approximation of the real-world statistics of appearances of the corresponding classes. The shot of a class $c$ is then set to:

$$k_c = \min \left\{ \lfloor R_c * (|\mathcal{S}| - |\mathcal{C}|) \rfloor + 1, |Im(c)| - q \right\}$$

which ensures that at least one example is selected for each class, with additional examples selected proportionally to $R_c$, if enough are available.

### .3   DATASETS

META-DATASET is formed of data originating from 10 different image datasets. A complete list of the datasets we use is the following.

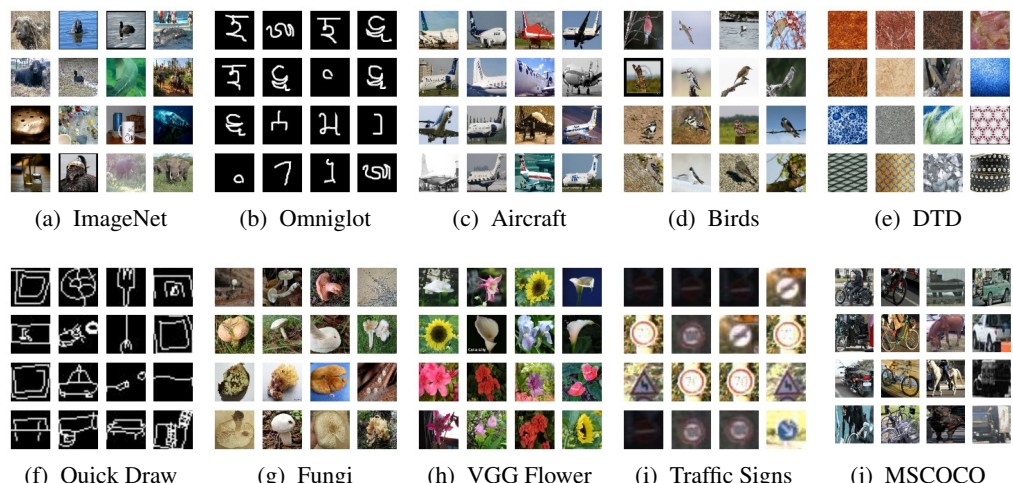

| (a) ImageNet | (b) Omniglot | (c) Aircraft | (d) Birds | (e) DTD |
| (f) Quick Draw | (g) Fungi | (h) VGG Flower | (i) Traffic Signs | (j) MSCOCO |

Figure 4:  Training examples taken from the various datasets forming META-DATASET.

**ILSVRC-2012 (ImageNet, Russakovsky et al., 2015)**   A dataset of natural images from 1000 categories (Figure 4a). We removed some images that were duplicates of images in another dataset in META-DATASET (43 images that were also part of Birds) or other standard datasets of interest (92 from Caltech-101 and 286 from Caltech-256). The complete list of duplicates is part of the source code release.

**Omniglot (Lake et al., 2015)**   A dataset of images of 1623 handwritten characters from 50 different alphabets, with 20 examples per class (Figure 4b). While recently Vinyals et al. (2016) proposed a

new split for this dataset, we instead make use of the original intended split Lake et al. (2015) which is more challenging since the split is on the level of alphabets (30 training alphabets and 20 evaluation alphabets), not characters from those alphabets, therefore posing a more challenging generalization problem. Out of the 30 training alphabets, we hold out the 5 smallest ones (i.e., with the least number of character classes) to form our validation set, and use the remaining 25 for training.

**Aircraft (Maji et al., 2013)**  A dataset of images of aircrafts spanning 102 model variants, with 100 images per class (Figure 4c). The images are cropped according to the providing bounding boxes, in order not to include other aircrafts, or the copyright text at the bottom of images.

**CUB-200-2011 (Birds, Wah et al., 2011)**  A dataset for fine-grained classification of 200 different bird species (Figure 4d). We did not use the provided bounding boxes to crop the images, instead the full images are used, which provides a harder challenge.

**Describable Textures (DTD, Cimpoi et al., 2014)**  A texture database, consisting of 5640 images, organized according to a list of 47 terms (categories) inspired from human perception (Figure 4e).

**Quick Draw (Jongejan et al., 2016)**  A dataset of 50 million black-and-white drawings across 345 categories, contributed by players of the game Quick, Draw! (Figure 4f).

**Fungi (Schroeder & Cui, 2018)**  A large dataset of approximately 100K images of nearly 1,500 wild mushrooms species (Figure 4g).

**VGG Flower (Nilsback & Zisserman, 2008)**  A dataset of natural images of 102 flower categories. The flowers chosen to be ones commonly occurring in the United Kingdom. Each class consists of between 40 and 258 images (Figure 4h).

**Traffic Signs (Houben et al., 2013)**  A dataset of 50,000 images of German road signs in 43 classes (Figure 4i).

**MSCOCO Lin et al. (2014)**  A dataset of images collected from Flickr with 1.5 million object instances belonging to 80 classes labelled and localized using bounding boxes. We choose the train2017 split and create images crops from original images using each object instance's groundtruth bounding box (Figure 4j).

### .4  HYPERPARAMETERS

We used three architectures: a commonly-used four-layer convolutional network, an 18-layer residual network and a wide residual network. While some of the baseline models performed best with the latter, we noticed that the meta-learners preferred the resnet-18 backbone and rarely the four-layer-convnet. For Relation Networks only, we also allow the option to use another architecture, aside from the aforementioned three, inspired by the four-layer-convnet used in the Relation Networks paper (Sung et al., 2018). The main difference is that they used the usual max-pooling operation only in the first two layers, omitting it in the last two, yielding activations of larger spatial dimensions. In our case, we found that these increased spatial dimensions did not fit in memory, so as a compromise we used max-pooling on the first 3 out of the 4 layer of the convnet.

For fo-MAML and fo-Proto-MAML, we tuned the inner-loop learning rate, the number of inner loop steps, and the number of additional such steps to be performed in evaluation (i.e., validation or test) episodes.

For the baselines, we tuned whether the cosine classifier of Baseline++ will be used, as opposed to a standard forward pass through a linear classification layer. Also, since Chen et al. (2019) added weight normalization (Salimans & Kingma, 2016) to their implementation of the cosine classifier layer, we also implemented this and created a hyperparameter choice for whether or not it is enabled. This hyperparameter is independent from the one that decides if the cosine classifier is used. Both are applicable to the $k$-NN Basline (for its all-way training classification task) and to the Finetune Baseline (both for its all-way training classification and for its within-episode classification

at validation and test times). For the Finetune Baseline, we tuned a binary hyperparameter deciding if gradient descent or ADAM is used for the within-task optimization. We also tuned the decision of whether all embedding layers are finetuned or, alternatively, the embedding is held fixed and only the final classifier on top of it is optimized. Finally, we tuned the number of finetuning steps that will be carried out.

We also tried two different image resolutions: the commonly-used 84x84 and 126x126. Finally, we tuned the learning rate schedule and weight decay and we used ADAM to train all of our models. All other details, dataset splits and the complete set of best hyperparameters discovered for each model are included in the source code.

## .5 COMPLETE MAIN RESULTS AND RANK COMPUTATION

**Rank computation**  We rank models by decreasing order of accuracy and handle ties by assigning tied models the average of their ranks. A tie between two models occurs when a 95% confidence interval statistical test on the difference between their mean accuracies is inconclusive in rejecting the null hypothesis that this difference is 0. Our recommendation is that this test is ran to determine if ties occur. As mentioned earlier, our source code will include this computation.

**Complete main tables**  For completeness, Table 2 presents a more detailed version of Table 1 that also displays confidence intervals and per-dataset ranks computed using the above procedure.

## .6 ANALYSIS OF PERFORMANCE ACROSS SHOTS AND WAYS

For completeness, in Figure 5 we show the results of the analysis of the robustness to different ways and shots for the variants of the models that were trained on all datasets. We observe the same trends as discussed in our Experiments section for the variants of the models that were trained on ImageNet.

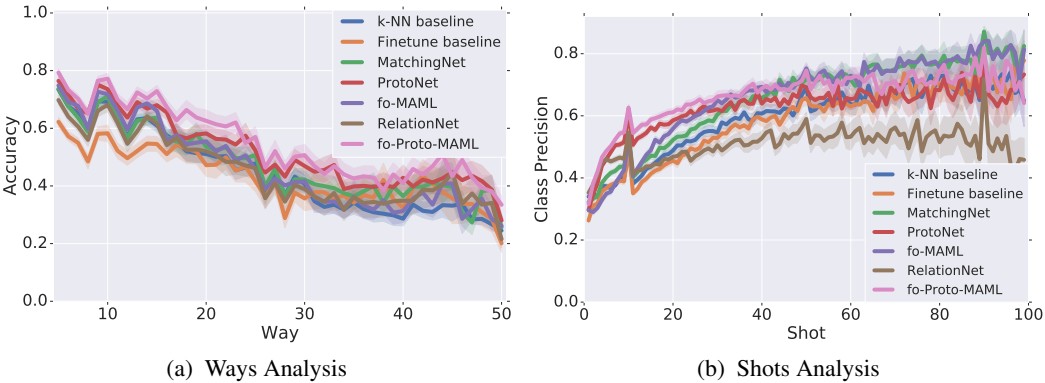

(a) Ways Analysis  (b) Shots Analysis

Figure 5:  Analysis of performance as a function of the episode's way, shots for models whose training source is (the training data of) all datasets. The bands display 95% confidence intervals.

## .7 EFFECT OF TRAINING ON ALL DATASETS OVER TRAINING ON ILSVRC-2012 ONLY

For more clearly observing whether training on all datasets leads to improved generalization over training on ImageNet only, Figure 6 shows side-to-side the performance of each model trained on ILSVRC only vs. all datasets. The difference between the performance of the all-dataset trained models versus the ImageNet-only trained ones is also visualized in Figure 1 in the main paper.

As discussed in the main paper, we notice that we do not always observe a clear generalization advantage in training from a wider collection of image datasets. While some of the datasets that were added to the meta-training phase did see an improvement across all models, in particular for Omniglot and Quick Draw, this was not true across the board. In fact, in certain cases the performance is slightly worse. We believe that more successfully leveraging diverse sources of data is an interesting open research problem.

Table 2: Few-shot classification results on META-DATASET.

(a) Models trained on **ILSVRC-2012 only**.

| Test Source | Method: Accuracy (%) ± confidence (%) | | | | | | |
|---|---|---|---|---|---|---|---|
| | $k$-NN | Finetune | MatchingNet | ProtoNet | fo-MAML | RelationNet | Proto-MAML |
| ILSVRC | 41.03±1.01 (6) | 45.78±1.10 (4) | 45.00±1.10 (4) | **50.50**±1.08 (1.5) | 45.51±1.11 (4) | 34.69±1.01 (7) | **49.53**±1.05 (1.5) |
| Omniglot | 37.07±1.15 (7) | 60.85±1.58 (2.5) | 52.27±1.28 (5) | 59.98±1.35 (2.5) | 55.55±1.54 (4) | 45.35±1.36 (6) | **63.37**±1.33 (1) |
| Aircraft | 46.81±0.89 (6) | **68.69**±1.26 (1) | 48.97±0.93 (5) | 53.10±1.00 (4) | 56.24±1.11 (2.5) | 40.73±0.83 (7) | 55.95±0.99 (2.5) |
| Birds | 50.13±1.00 (6.5) | 57.31±1.26 (5) | 62.21±0.95 (3.5) | **68.79**±1.01 (1.5) | 63.61±1.06 (3.5) | 49.51±1.05 (6.5) | **68.66**±0.96 (1.5) |
| Textures | 66.36±0.75 (4) | **69.05**±0.90 (1.5) | 64.15±0.85 (6) | 66.56±0.83 (4) | **68.04**±0.81 (1.5) | 52.97±0.69 (7) | 66.49±0.83 (4) |
| Quick Draw | 32.06±1.08 (7) | 42.60±1.17 (4.5) | 42.87±1.09 (4.5) | 48.96±1.08 (2) | 43.96±1.29 (4.5) | 43.30±1.08 (4.5) | **51.52**±1.00 (1) |
| Fungi | 36.16±1.02 (4) | 38.20±1.02 (3) | 33.97±1.00 (5) | **39.71**±1.11 (1.5) | 32.10±1.10 (6) | 30.55±1.04 (7) | **39.96**±1.14 (1.5) |
| VGG Flower | 83.10±0.68 (4) | 85.51±0.68 (2.5) | 80.13±0.71 (6) | 85.27±0.77 (2.5) | 81.74±0.83 (5) | 68.76±0.83 (7) | **87.15**±0.69 (1) |
| Traffic Signs | 44.59±1.19 (6) | **66.79**±1.31 (1) | 47.80±1.14 (3.5) | 47.12±1.10 (5) | 50.93±1.51 (2) | 33.67±1.05 (7) | 48.83±1.09 (3.5) |
| MSCOCO | 30.38±0.99 (6.5) | 34.86±0.97 (4) | 34.99±1.00 (4) | 41.00±1.10 (2) | 35.30±1.23 (4) | 29.15±1.01 (6.5) | **43.74**±1.12 (1) |
| **Avg. rank** | 5.7 | 2.9 | 4.65 | 2.65 | 3.7 | 6.55 | **1.85** |

(b) Models trained on **all datasets**.

| Test Source | Method: Accuracy (%) ± confidence (%) (rank) | | | | | | |
|---|---|---|---|---|---|---|---|
| | $k$-NN | Finetune | MatchingNet | ProtoNet | fo-MAML | RelationNet | Proto-MAML |
| ILSVRC | 38.55±0.94 (4.5) | 43.08±1.08 (2.5) | 36.08±1.00 (6) | 44.50±1.05 (2.5) | 37.83±1.01 (4.5) | 30.89±0.93 (7) | **46.52**±1.05 (1) |
| Omniglot | 74.60±1.08 (6) | 71.11±1.37 (7) | 78.25±1.01 (4.5) | 79.56±1.12 (4.5) | 83.92±0.95 (2.5) | **86.57**±0.79 (1) | 82.69±0.97 (2.5) |
| Aircraft | 64.98±0.82 (7) | 72.03±1.07 (3.5) | 69.17±0.96 (5.5) | 71.14±0.86 (3.5) | **76.41**±0.69 (1) | 69.71±0.83 (5.5) | 75.23±0.76 (2) |
| Birds | 66.35±0.92 (2.5) | 59.82±1.15 (5) | 56.40±1.00 (6) | 67.01±1.02 (2.5) | 62.43±1.08 (4) | 54.14±0.99 (7) | **69.88**±1.02 (1) |
| Textures | 63.58±0.79 (5) | **69.14**±0.85 (1.5) | 61.80±0.74 (6) | 65.18±0.84 (3.5) | 64.14±0.83 (3.5) | 56.56±0.73 (7) | **68.25**±0.81 (1.5) |
| Quick Draw | 44.88±1.05 (7) | 47.05±1.16 (6) | 60.81±1.03 (3.5) | 64.88±0.89 (2) | 59.73±1.10 (5) | 61.75±0.97 (3.5) | **66.84**±0.94 (1) |
| Fungi | 37.12±1.06 (3.5) | 38.16±1.04 (3.5) | 33.70±1.04 (6) | 40.26±1.13 (2) | 33.54±1.11 (6) | 32.56±1.08 (6) | **41.99**±1.17 (1) |
| VGG Flower | 83.47±0.61 (4) | 85.28±0.69 (3) | 81.90±0.72 (5) | 86.85±0.71 (2) | 79.94±0.84 (6) | 76.08±0.76 (7) | **88.72**±0.67 (1) |
| Traffic Signs | 40.11±1.10 (6) | **66.74**±1.23 (1) | 55.57±1.08 (2) | 46.48±1.00 (4) | 42.91±1.31 (5) | 37.48±0.93 (7) | 52.42±1.08 (3) |
| MSCOCO | 29.55±0.96 (5) | 35.17±1.08 (3) | 28.79±0.96 (5) | 39.87±1.06 (2) | 29.37±1.08 (5) | 27.41±0.89 (7) | **41.74**±1.13 (1) |
| **Avg. rank** | 5.05 | 3.6 | 4.95 | 2.85 | 4.25 | 5.8 | **1.5** |

Figure 6: Accuracy on the test datasets, when training on ILSVRC only or All datasets (same results as shown in the main tables). The bars display 95% confidence intervals.

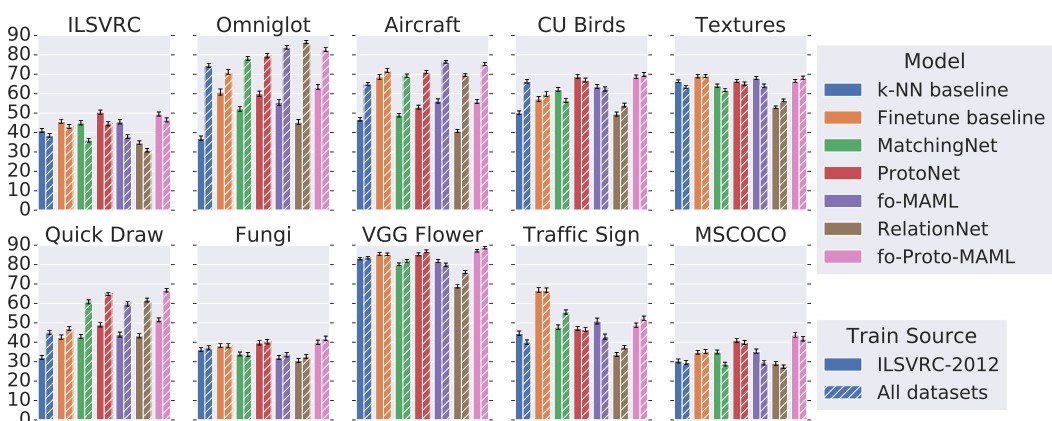

## .8 EFFECT OF PRE-TRAINING VERSUS TRAINING FROM SCRATCH

For each meta-learner, we selected the best model (based on validation on ImageNet's validation split) out of the ones that used the pre-trained initialization, and the best out of the ones that trained from scratch. We then ran the evaluation of each on (the test split of) all datasets in order to quantify how beneficial this pre-trained initialization is. We performed this experiment twice: for the models that are trained on ImageNet only and for the models that are trained on (the training splits of) all datasets.

The results of this investigation were reported in the main paper in Figure 3a and Figure 3b, for ImageNet-only training and all dataset training, respectively. We show the same results in Figure 7, printed larger to facilitate viewing of error bars. For easier comparison, we also plot the difference in performance of the models that were pre-trained over the ones that weren't, in Figures 8a and 8b. These figures make it easier to spot that while using the pre-trained solution usually helps for datasets that are visually not too different from ImageNet, it may hurt for datasets that are significantly different from it, such as Omniglot, Quickdraw (and surprisingly Aircraft). Note that these three datasets are the same three that we found benefit from training on All datasets instead of ImageNet-only. It appears that using the pre-trained solution biases the final solution to specialize on ImageNet-like datasets.

## .9 EFFECT OF META-LEARNING VERSUS INFERENCE-ONLY

Figure 9 shows the same plots as in Figures 3c and 3d but printed larger to facilitate viewing of error bars. Furthermore, as we have done for visualizing the observed gain of pre-training, we also present in Figures 10a and 10b the gain observed from meta-learning as opposed to training the corresponding inference-only baseline, as explained in the Experiments section of the main paper. This visulization makes it clear that while meta-training usually helps on ImageNet (or doesn't hurt too much), it sometimes hurts when it is performed on all datasets, emphasizing the need for further research into best practices of meta-learning across heterogeneous sources.

## .10 FINEGRAINEDNESS ANALYSIS

We investigate the hypothesis that finer-grained tasks are more challenging than coarse-grained ones by creating binary ImageNet episodes with the two classes chosen uniformly at random from the DAG's set of leaves. We then define the degree of coarse-grainedness of a task as the height of the lowest common ancestor of the two chosen leaves, where the height is defined as the length of the longest path from the lowest common ancestor to one of the selected leaves. Larger heights then correspond to coarser-grained tasks. We present these results in Figure 11. We do not detect a significant trend when performing this analysis on the test DAG. The results on the training DAG,

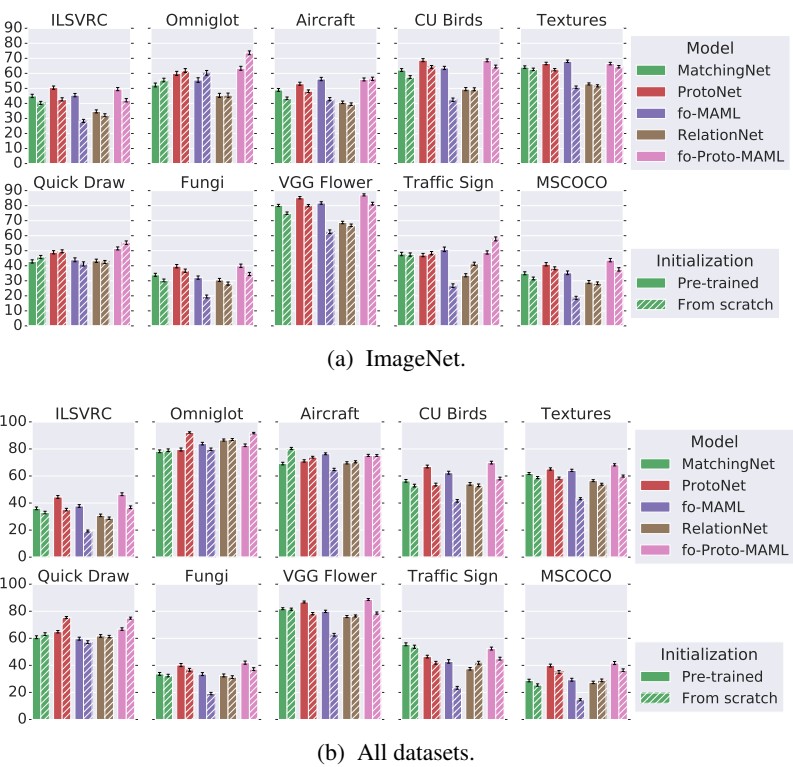

(a) ImageNet.

(b) All datasets.

Figure 7: Comparing pre-training to starting from scratch. Same plots as Figure 3a and Figure 3b, only larger.

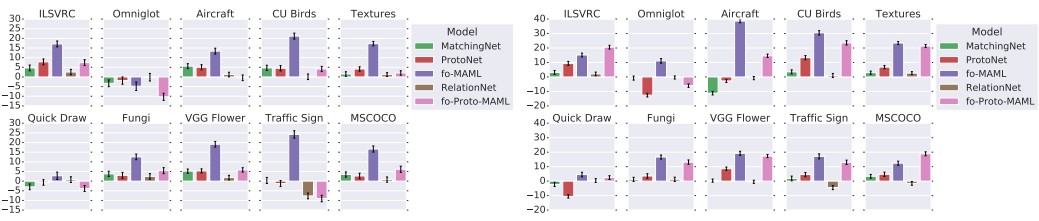

(a) The gain from pre-training (ImageNet).     (b) The gain from pre-training (All datasets).

Figure 8: The performance difference of initializing the embedding weights from a pre-trained solution, before episodically training on ImageNet or all datasets, over using a random initialization of those weights. The pre-trained weights that we consider are the ones that the $k$-NN baseline converged to when it was trained on ImageNet. Positive values indicate that this pre-training was beneficial.

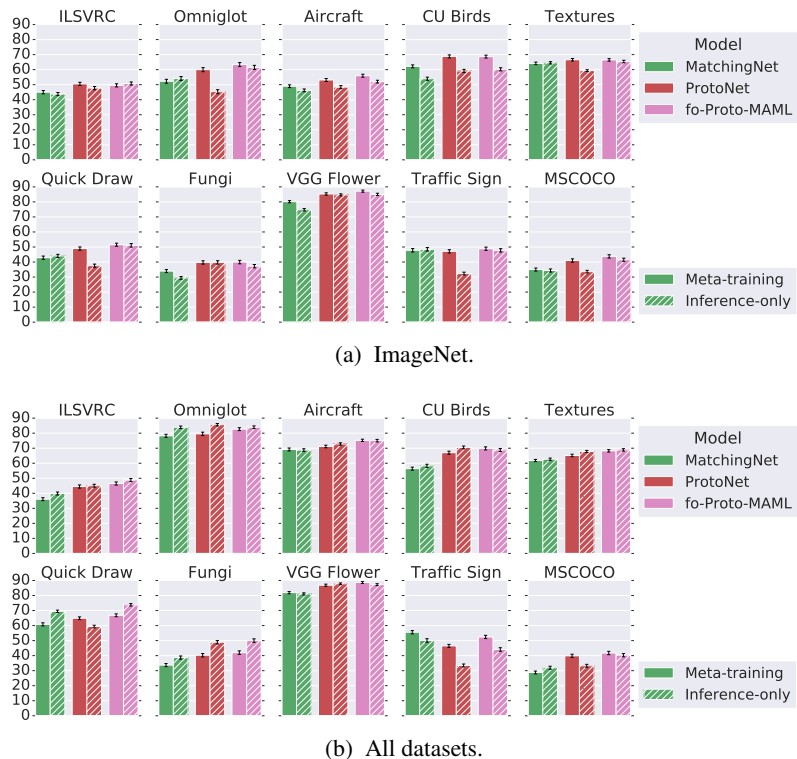

(a) ImageNet.

(b) All datasets.

Figure 9: Comparing the meta-trained variant of meta-learners against their inference-only counterpart. Same plots as Figure 3c and Figure 3d, only larger.

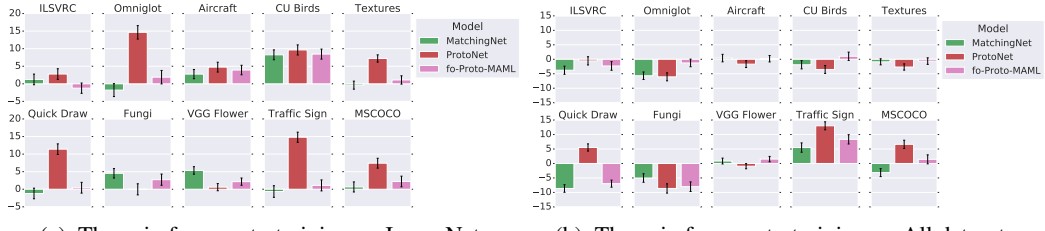

(a) The gain from meta-training on ImageNet.      (b) The gain from meta-training on All datasets.

Figure 10: The performance difference of meta-learning over the corresponding inference-only baseline of each meta-learner. Positive values indicate that meta-learning was beneficial.

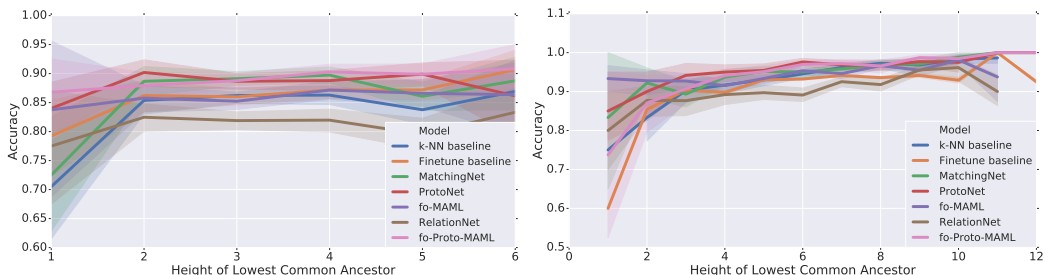

(a) Fine-grainedness Analysis (on ImageNet's test (b) Fine-grainedness Analysis (on ImageNet's train graph)      graph graph)

Figure 11: Analysis of performance as a function of the degree of fine-grainedness. Larger heights correspond to coarser-grained tasks. The bands display 95% confidence intervals.

Table 3: Improvement of fo-MAML when using a larger inner learning rate $\alpha$.

(a) Models trained on **ILSVRC-2012 only**.

| Test Source | Method: Accuracy (%) ± confidence (%) | |
| --- | --- | --- |
| | fo-MAML $\alpha = 0.01$ (old) | fo-MAML $\alpha \approx 0.1$ |
| ILSVRC | 36.09±1.01 | 45.51±1.11 |
| Omniglot | 38.67±1.39 | 55.55±1.54 |
| Aircraft | 34.50±0.90 | 56.24±1.11 |
| Birds | 49.10±1.18 | 63.61±1.06 |
| Textures | 56.50±0.80 | 68.04±0.81 |
| Quick Draw | 27.24±1.24 | 43.96±1.29 |
| Fungi | 23.50±1.00 | 32.10±1.10 |
| VGG Flower | 66.42±0.96 | 81.74±0.83 |
| Traffic Signs | 33.23±1.34 | 50.93±1.51 |
| MSCOCO | 27.52±1.11 | 35.30±1.23 |

(b) Models trained on **all datasets**.

| Test Source | Method: Accuracy (%) ± confidence (%) | |
| --- | --- | --- |
| | fo-MAML $\alpha = 0.01$ (old) | fo-MAML $\alpha \approx 0.1$ |
| ILSVRC | 32.36±1.02 | 37.83±1.01 |
| Omniglot | 71.91±1.20 | 83.92±0.95 |
| Aircraft | 52.76±0.90 | 76.41±0.69 |
| Birds | 47.24±1.14 | 62.43±1.08 |
| Textures | 56.66±0.74 | 64.16±0.83 |
| Quick Draw | 50.50±1.19 | 59.73±1.10 |
| Fungi | 21.02±0.99 | 33.54±1.11 |
| VGG Flower | 70.93±0.99 | 79.94±0.84 |
| Traffic Signs | 34.18±1.26 | 42.91±1.31 |
| MSCOCO | 24.05±1.10 | 29.37±1.08 |

though, do seem to indicate that our hypothesis holds to some extent. We conjecture that this may be due to the richer structure of the training DAG, but we encourage further investigation.

### .11   THE IMPORTANCE OF MAML'S INNER-LOOP LEARNING RATE HYPERPARAMETER.

The camera-ready version includes updated results for MAML and Proto-MAML following an external suggestion to experiment with larger values for the inner-loop learning rate $\alpha$ of MAML. We found that re-doing our hyperparameter search with a revised range that includes larger $\alpha$ values significantly improved fo-MAML's performance on META-DATASET. For consistency, we applied the same change to fo-Proto-MAML and re-ran those experiments too.

We found that the value of this $\alpha$ that performs best for fo-MAML both for training on ImageNet only and training on all datasets is approximately 0.1, which is an order of magnitude larger than our previous best value. Interestingly, fo-Proto-MAML does not choose such a large $\alpha$ value, with best $\alpha$ being 0.0054 when training on ImageNet only and 0.02 when training on all datasets. Plausibly this difference can be attributed to the better initialization of Proto-MAML which requires a less aggressive optimization for the adaptation to each new task. This hypothesis is also supported by the fact that fo-Proto-MAML chooses to take fewer adaptation steps than fo-MAML does. The complete set of best discovered hyperparameters is available in our public code.

To emphasize the importance of properly tuning this hyperparameter, Table 3 displays the previous best and the new best fo-MAML results side-by-side, showcasing the large performance gap when using the appropriate value for $\alpha$.

### .12 The choice of a meta-validation procedure for Meta-Dataset

The design choice we made, as discussed in the main paper, is to use (the meta-validation set of) ImageNet only for model selection in all of our experiments. In the absence of previous results on the topic, this is a reasonable strategy since ImageNet has been known to consitute a useful proxy for performance on other datasets. However, it is likely that this is not the optimal choice: there might be certain hyperparameters for a given model that work best for held-out ImageNet episodes, but not for held-out episodes of other datasets. An alternative meta-validation scheme would be to use the average (across datasets) validation accuracy as an indicator for early stopping and model selection. We did not choose this method due to concerns about the reliability of this average performance. Notably, taking a simple average would over-emphasize larger datasets, or might over-emphasize datasets with natural images (as opposed to Omniglot and Quickdraw). Nevertheless, whether this strategy is beneficial is an interesting empirical question.

### .13 Additional Per-dataset Analysis of Shots and Ways

In our previous analysis of performance across different shots and ways (Figures 2b, 2a, and 5), the performance is averaged over all evaluation datasets. In this section we further break down those plots by presenting the results separately for each dataset. Figures 12 and 13 show the analysis of performance as a function of ways and shots (respectively) for each evaluation dataset, for models that were trained on ImageNet only. For completeness, Figures 14 and 15 show the same for the models trained on (the training splits of) all datasets.

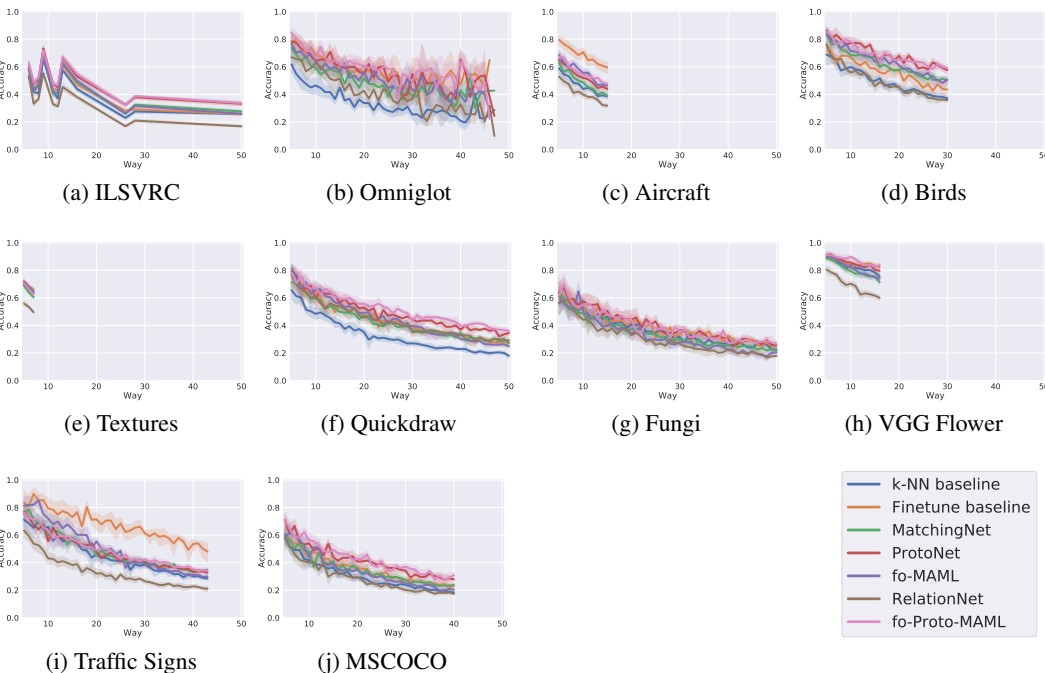

Figure 12: The performance across different ways, with 95% confidence intervals, shown separately for each evaluation dataset. All models had been trained on ImageNet-only.

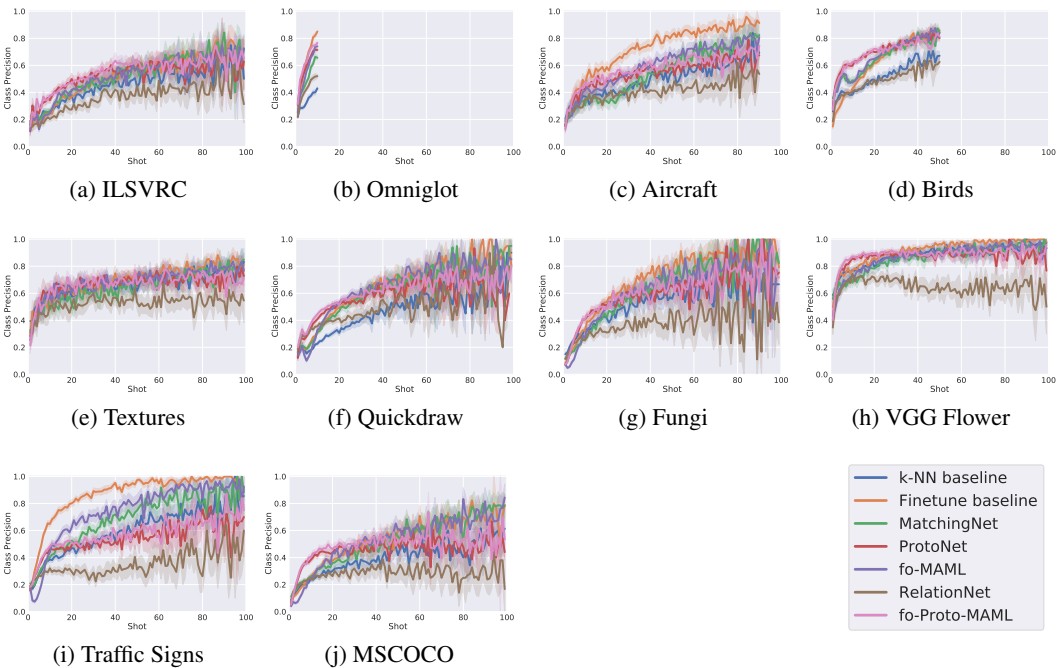

Figure 13: The performance across different shots, with 95% confidence intervals, shown separately for each evaluation dataset. All models had been trained on ImageNet-only.

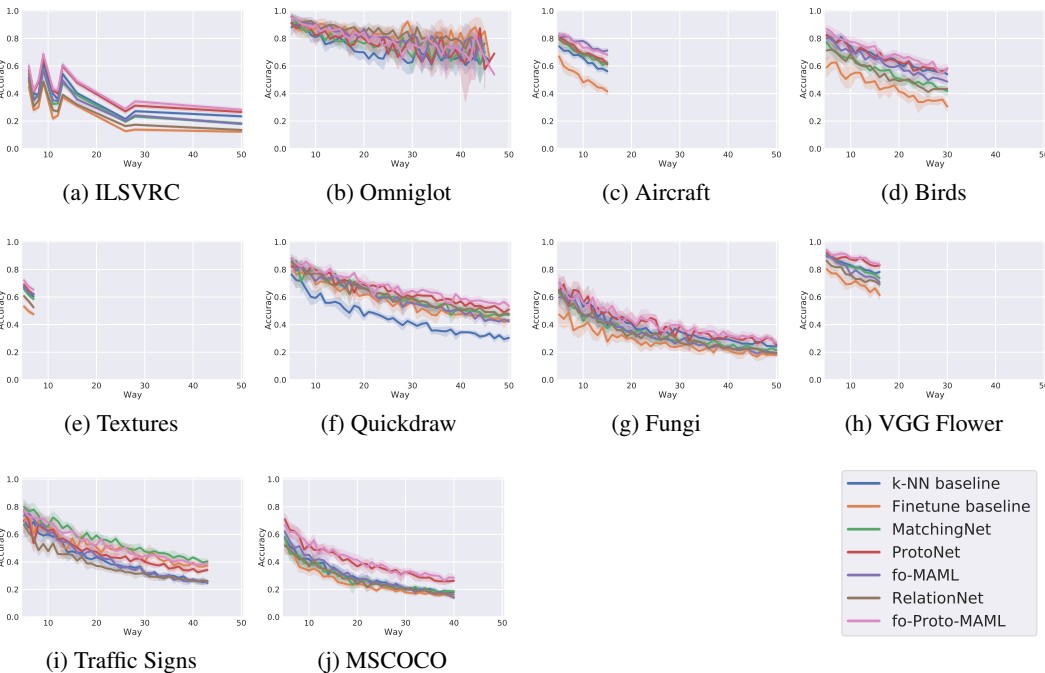

Figure 14: The performance across different ways, with 95% confidence intervals, shown separately for each evaluation dataset. All models had been trained on (the training splits of) all datasets.

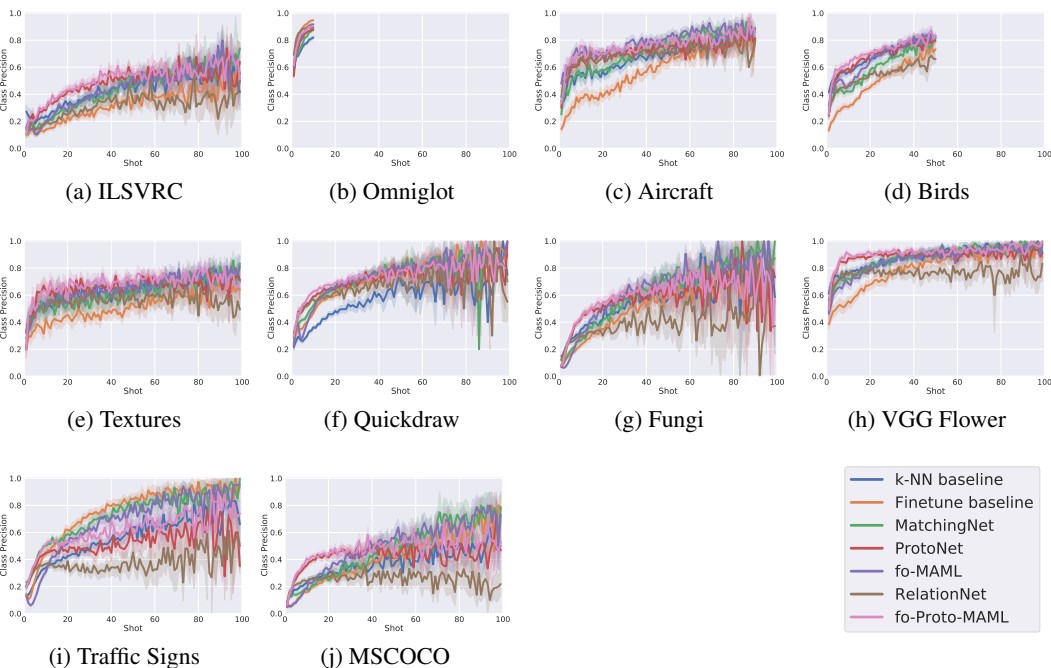

Figure 15: The performance across different shots, with 95% confidence intervals, shown separately for each evaluation dataset. All models had been trained on (the training splits of) all datasets.

