# OpenReview forum: "Meta-Dataset: A Dataset of Datasets for Learning to Learn from Few Examples"
_ICLR.cc/2020/Conference — Accept (Poster)_

### Official Review · AnonReviewer2 · 2019-10-18
**Official Blind Review #2**

**Rating:** 8

**Review:**

The paper presents Meta-Dataset, a benchmark for few-shot classification that combines various image classification data sets, allows the number of classes and examples per class to vary, and considers the relationships between classes. It performs an empirical evaluation of six algorithms from the literature, k-NN, FineTune, Prototypical Networks, Matching Networks, Relation Networks, and MAML. A new approach combining Prototypical Networks and first-order MAML is shown to outperform those algorithms, but there is substantial room for improvement overall.

Many papers in the literature have observed that current algorithms achieve very high performance on existing few-shot classification data sets such as Omniglot, making it difficult to compare them. This paper fills the need for a more challenging data set. This data set is also more realistic in that
1. The training and test data may come from different data sets. In the real world, we often encounter locations, weather conditions, etc. that are unseen during training, and agents must be able to adapt quickly to these new situations.
2. Classes are selected based on their semantic structure. This allows control of the difficulty of the task, as it is easier to adapt to a semantically similar class, and thus enables a more nuanced comparison of few-shot classification algorithms.

Suggestions for clarifying the writing:
1. Section 3 should also discuss choosing the data set to sample from.
2. In figures 1c and 1d, it would be helpful to include standard error regions, like in the rest of figure 1.
3. Maybe the paragraph on Proto-MAML should be moved to section 3, as it is not from previous literature. In addition, steps 2b and 2c in section 3.2 overlap in content, and so it may be clearer to combine them.

**Experience Assessment:**

I have read many papers in this area.

**Review Assessment: Checking Correctness Of Derivations And Theory:**

I assessed the sensibility of the derivations and theory.

**Review Assessment: Checking Correctness Of Experiments:**

I assessed the sensibility of the experiments.

**Review Assessment: Thoroughness In Paper Reading:**

I read the paper at least twice and used my best judgement in assessing the paper.

---

> ### Author Response · Authors · 2019-11-09
> **Response to Review #2**
>
> We thank Reviewer 2 for their constructive feedback and suggestions! Our responses are below.
>
> 1. The steps in our Episode Sampling section assume that we are “given a particular split of a particular dataset”. But we agree it would be clearer to only assume we are given a particular split, and add “Step 0: Uniformly sample a dataset from the set of datasets that have classes assigned to the chosen split”. We will update the text to reflect this suggestion.
> 2. Good idea, we will update Figure 1c,d to also include error bars.
> 3. We agree it seems odd to present Proto-MAML in the Background section, as it is one of the contributions of this work, and not background. Section 3 however is dedicated to the description of the Meta-Dataset benchmark, so we didn’t want to blend that with describing models. Perhaps it would be clearer to rename the Background section to something like: “Few-shot Classification Task and Approaches”. Alternatively, we can create a separate sub-section or section solely for introducing Proto-MAML. We thank Reviewer 2 for pointing this out and look forward to hearing their thoughts on this.
> 4. Steps 2b and 2c describe two different processes: 2b computes the *total* support set size, and 2c then decides how to distribute that total budget among the different classes that will participate in the episode. We therefore believe it’s clearer to keep them separate, but we’re happy to discuss this further if Reviewer 2 still disagrees.

---

### Official Review · AnonReviewer1 · 2019-10-24
**Official Blind Review #1**

**Rating:** 6

**Review:**

This paper proposed a really interesting direction for few-shot and meta-learning, the concept of a 'meta-dataset', which can be used for more realistic evaluation of algorithms. The main contributions are:

1) A more realistic environment for training and testing few-shot learners.
2) Experimental evaluation of popular models
3) Analyses of whether different models benefit from more data,
4) A new meta-learner

I think this work is an interesting empirical paper which should be supported by solid experimental results. My concern about this paper in its current form is that the layout/structure of the paper needs to be improved, for example:

Considering putting some of the key results in the appendix section in the main text
Removal of repeating results from the main text by shortening the early sections

**Experience Assessment:**

I have published one or two papers in this area.

**Review Assessment: Checking Correctness Of Derivations And Theory:**

I assessed the sensibility of the derivations and theory.

**Review Assessment: Checking Correctness Of Experiments:**

I assessed the sensibility of the experiments.

**Review Assessment: Thoroughness In Paper Reading:**

I read the paper at least twice and used my best judgement in assessing the paper.

---

> ### Author Response · Authors · 2019-11-09
> **Response to Review #1**
>
> We thank Reviewer 1 for their constructive feedback!
>
> We agree that it would be great to move some of the plots from the Appendix in the main text. We tried our best to keep the submission within the recommended 8 page limit while at the same time including most important results in the main paper. If all reviewers agree though, we would love to use some extra space (the hard limit is 10 pages if we understand correctly) to move some of the figures from the Appendix into the main paper.
>
> Meanwhile, we tried our best to arrange things so that the Appendix does not have too many new results. Instead, most of the plots there offer different viewpoints of the results already presented in the main paper. More concretely, in the Appendix:
> - Figure 3 performs the same analysis as Figure 1c,d but for the case of training on all datasets. This is presented for completeness but the conclusion is the same as for the ImageNet-only case which is presented in the main paper.
> - Figure 4 is an alternative visualization of Table 1 in bar plot form (there is no new information there).
> - Figure 5 is computed by taking the element-wise difference of the two sets of results in Table 1 (trained on all datasets - trained on ImageNet only). It offers a perhaps more convenient way of visualizing those results but also does not contain any new information that is not in Table 1.
> - Figures 6 and 8 contain exactly the same plots as Figure 1a,b, and Figure 1e,f, respectively, only printed larger for easier viewing of error bars.
> - Figures 7 and 9 offer alternative views of the barplots in Figures 6 and 8, respectively (do not contain any additional information).
> - Table 2 contains the same results as Table 1 with the only difference of displaying confidence intervals and per-dataset ranks too, for completeness.
> - Figure 10 contains the fine-grainedness analysis that we discuss in the main text just above the conclusion. This is not a main result so we think that simply mentioning the take-away in the main paper is sufficient.
>
> Regarding the second suggestion: “Removal of repeating results from the main text”, we were not sure which “repeating results” Reviewer 1 is referring to and we’d appreciate a clarification, so we can appropriately take it into consideration.

---

### Official Review · AnonReviewer3 · 2019-10-26
**Official Blind Review #3**

**Rating:** 3

**Review:**

The authors of this paper construct a new few-shot learning dataset. The whole dataset consists of several data from different sources. The authors test several representative meta-learning models (e.g., matching network, Prototype network, MAML) on this dataset and give the analysis. Furthermore, the authors combine MAML and Prototype network, which achieves the best performance on this new dataset.

Pros:
+ Compared with previous datasets (e.g., miniimagenet, omniglot), the constructed meta-dataset is larger and more realistic, which contains several datasets collected from different sources
+ Several competitive baselines are compared on this dataset under different scenarios (e.g., different number of shot) with reasonable analysis.

Cons:
- I am familiar with meta-learning, however, it is my first time to review a paper whose main contribution is proposing a new benchmark. The proposed dataset may useful in further meta-learning research. However, I do not feel the construction way is quite difficult. The authors only propose several rules to construct the data set (see 3.2).
- It would be better if the authors can explain more about Proto-MAML. My understanding of Proto-MAML is to apply the prototype on the last layer and keep the other layers.

-------------------------------------------------------------------------------------------------------------------------------------------------
After rebuttal:

After reading the response, I think constructing a new benchmark is important and useful. However, considering the technical contributions of this paper, I finally decide to keep my score.


**Experience Assessment:**

I have published one or two papers in this area.

**Review Assessment: Checking Correctness Of Derivations And Theory:**

I assessed the sensibility of the derivations and theory.

**Review Assessment: Checking Correctness Of Experiments:**

I assessed the sensibility of the experiments.

**Review Assessment: Thoroughness In Paper Reading:**

I read the paper at least twice and used my best judgement in assessing the paper.

---

> ### Author Response · Authors · 2019-11-09
> **Response to Review #3**
>
> We’d like to thank Reviewer 3 for their comments!
>
> We kindly ask the reviewer to clarify what they mean by “However, I do not feel the construction way is quite difficult”. As far as we’re concerned, the aim of a benchmark is not to be complicated, but to serve as a useful proxy for the task of interest. Our goal with Meta-Dataset is to introduce an environment for examining a more realistic variant of few-shot classification, including underemphasized aspects such as variable shots and ways, class imbalance, class structure, and heterogeneous data.
>
> Proto-MAML, as explained in the paper, performs a simple modification over the original MAML algorithm: the linear classification layer for each task is initialized from the prototypical layer (see the “Introducing Proto-MAML” paragraph for the precise formulation). This is the only difference compared to vanilla MAML. Intuitively then, the aim is to meta-learn the embedding weights such that: given a new task, initializing the output layer from the prototypes of that task and performing a few adaptation steps on the embedding and output layer suffice for performing well on the query set of that task.
>
> We’d be happy to answer any other questions about Proto-MAML and update the text accordingly if Reviewer 3 thinks our explanation is unclear. We have also attached our code with the submission to clarify any additional details about the method.

---

> > ### Author Response · Authors · 2019-11-14
> > **We'd love to hear back from Reviewer 3**
> >
> > We would love to hear back from Reviewer 3 about whether our explanation of Proto-MAML is still unclear, so we can appropriately take their feedback into account.
> > It would also be great to hear whether Reviewer 3 still has concerns about the lack of difficulty of construction of our benchmark. We would appreciate the opportunity to discuss this further if our response has not already convinced the reviewer.

---

> > > ### Comment · AnonReviewer3 · 2019-11-15
> > > **Feedback to the authors' response**
> > >
> > > Dear Authors,
> > >
> > > I appreciate your response. I've understood your contributions to constructing this dataset. For Proto-MAML, my initial understanding is correct. Thank you for clarification. I will read your revised version later.

---

### Author Response · Authors · 2019-11-13
**Revision to address reviewer feedback**

We have uploaded a revised version of the submission, with the following changes:

- In Section 3, we now explicitly mention the uniform choice of dataset as a step of the episode sampling, as per Reviewer 2’s first suggestion.
- Added confidence intervals to all figures, as per Reviewer 2’s second suggestion.
- Renamed the “Background” section to “Few-shot Classification: Task Formulation and Approaches” (based on our response to Reviewer 2’s third suggestion — subject to change).
- Slight changes to the results of Finetune Baseline, due to this model having converged to a slightly better solution shortly after the deadline. The changes are small and only affect this model.
- Added (in the Appendix) per-evaluation-dataset plots of the analysis of ways and shots in case this is of interest, since the main analyses we present for this aggregate the results over all evaluation datasets.
- Changed the colors in all figures so that a given model always receives the same color across figures. Also changed the order of the datasets in the barplots (Figure 1a, 1b, 1e, 1f, and related ones in the Appendix) to agree with the order in which the datasets appear in Tables 1 and 2.
- Fixed a few typos and added minor clarifications.

We will continue to revise the paper based on all Reviewers’ responses to our rebuttal.

---

### Decision · Program_Chairs · 2019-12-19

**Decision:**

Accept (Poster)

**Comment:**

While the reviewers have some outstanding issues regarding the organization and clarity of the paper, the overall consensus is that the proposed evaluation methods is a useful improvement over current standards for meta-learning.